# Axonal Regeneration: Underlying Molecular Mechanisms and Potential Therapeutic Targets

**DOI:** 10.3390/biomedicines10123186

**Published:** 2022-12-08

**Authors:** Rabia Akram, Haseeb Anwar, Muhammad Shahid Javed, Azhar Rasul, Ali Imran, Shoaib Ahmad Malik, Chand Raza, Ikram Ullah Khan, Faiqa Sajid, Tehreem Iman, Tao Sun, Hyung Soo Han, Ghulam Hussain

**Affiliations:** 1Neurochemicalbiology and Genetics Laboratory (NGL), Department of Physiology, Faculty of Life Sciences, Government College University, Faisalabad 38000, Pakistan; 2Department of Physiology, Sargodha Medical College, Sargodha 40100, Pakistan; 3Department of Zoology, Faculty of Life Sciences, Government College University, Faisalabad 38000, Pakistan; 4Department of Food Sciences, Government College University, Faisalabad 38000, Pakistan; 5Department of Biochemistry, Sargodha Medical College, Sargodha 40100, Pakistan; 6Department of Zoology, Faculty of Chemistry and Life Sciences, Government College University, Lahore 54000, Pakistan; 7Department of Pharmaceutics, Faculty of Pharmaceutical Sciences, Government College University, Faisalabad 38000, Pakistan; 8Center for Precision Medicine, School of Medicine and School of Biomedical Sciences, Huaqiao University, Xiamen 361021, China; 9Department of Physiology, School of Medicine, Clinical Omics Institute, Kyungpook National University, Daegu 41944, Republic of Korea

**Keywords:** axonal regeneration, nerve injury, regeneration-associated genes, neurotrophic factors, cyclic adenosine monophosphate, microRNAs

## Abstract

Axons in the peripheral nervous system have the ability to repair themselves after damage, whereas axons in the central nervous system are unable to do so. A common and important characteristic of damage to the spinal cord, brain, and peripheral nerves is the disruption of axonal regrowth. Interestingly, intrinsic growth factors play a significant role in the axonal regeneration of injured nerves. Various factors such as proteomic profile, microtubule stability, ribosomal location, and signalling pathways mark a line between the central and peripheral axons’ capacity for self-renewal. Unfortunately, glial scar development, myelin-associated inhibitor molecules, lack of neurotrophic factors, and inflammatory reactions are among the factors that restrict axonal regeneration. Molecular pathways such as cAMP, MAPK, JAK/STAT, ATF3/CREB, BMP/SMAD, AKT/mTORC1/p70S6K, PI3K/AKT, GSK-3β/CLASP, BDNF/Trk, Ras/ERK, integrin/FAK, RhoA/ROCK/LIMK, and POSTN/integrin are activated after nerve injury and are considered significant players in axonal regeneration. In addition to the aforementioned pathways, growth factors, microRNAs, and astrocytes are also commendable participants in regeneration. In this review, we discuss the detailed mechanism of each pathway along with key players that can be potentially valuable targets to help achieve quick axonal healing. We also identify the prospective targets that could help close knowledge gaps in the molecular pathways underlying regeneration and shed light on the creation of more powerful strategies to encourage axonal regeneration after nervous system injury.

## 1. Introduction

Axonal loss is one of the most common and severe symptoms of brain and spinal cord injuries. Axonal injury is triggered by various neurotoxins and neurological illnesses, which result in the loss of neural connections and the failure of axonal regeneration [1,2]. The capacity of a neuron to regenerate, fortunately, plays a vital role in the healing of severed axons. The rate of axonal regeneration is affected by several factors, including the intrinsic growth capacity of neurons, the rate of protein synthesis, cytoskeletal organization, rapid or slow clearance of myelin debris from a damaged nerve, along with the optimal growth factor supplies [3,4].

Due to the self-repairing ability of peripheral neurons and the quick reactivation of intrinsic growth programs, axons in the peripheral nervous system (PNS) can regenerate spontaneously after damage. The distal part of an axon is separated from the neuronal cell body after injury, resulting in Wallerian degeneration, which causes disintegration and fragmentation of the axon. The clearance of debris by glial cells, particularly macrophages, is a key component of PNS axonal regeneration and prevents the formation of scars that obstruct the process of axonal regeneration. This leads to the active regeneration of the proximal stump of the axon and successful target reinnervation. Immediately after a peripheral nerve injury, cytoskeletal modifications trigger cascades of events that promote axonal outgrowth and growth cone formation. Later, injury activates transcription factors that further turn on pro-regenerative transcriptional programs and facilitate axon rejuvenation [5,6]. However, in the adult central nervous system (CNS), axonal abrasion causes a reduction in the regeneration ability of axons due to a lack of intrinsic growth factors [7]. Even in CNS injury, various signalling pathways, including mTOR-S6, cAMP, and JAK-STAT3, are activated and enhance the process of regeneration in stark contrast to peripheral nerve injury [4,8].

Apart from the regenerative potential of both peripheral and central axons, researchers are now working to understand the molecular pathways that activate the regenerative program for axonal regeneration in mammalian PNS and CNS [9]. Identification and modulation of these pathways may give therapeutic approaches to improve neuronal functional recovery after axonal damage.

## 2. CNS vs. PNS Regeneration

Abortive and unsuccessful axonal regeneration in the CNS is partially ascribed to the altered proteomic profile of the CNS, particularly the involvement of the extracellular milieu around the damaged axons. Upon injury, oligodendrocyte precursor cells, reactive astrocytes, and reactive macrophages migrate toward the lesion site and release myelin-associated inhibitors, chondroitin sulfate proteoglycan (CSPG), and cytokines that create an inhibitory environment for CNS axons growth. Further, the lesion contributes to forming trabeculated or cystic cavities, due to which axons lose their attachment to the lesion site for growth. In the PNS, robust axonal regeneration occurs after injury since none of the inhibitory molecules mentioned above is present [10].

Microtubule stabilization is the second distinction between central and peripheral axons. After an injury, most CNS axons pull back and allow minimal sprouting of axons only for a short distance. Dystrophic and swollen endings of sprouted axons fail to start the stabilization of microtubules in the growth cones due to their exposure to an inhibitory environment of CNS. According to previous research, retraction bulbs of the CNS exhibit disorderly microtubules in contrast to the organized bundles of microtubules found in growth cones of the PNS [11]. The absence of ribosomes and mRNAs in CNS axons is a third mechanism in limiting regeneration. Successful regeneration of peripheral nerves has been intimately linked to mRNA localization and protein synthesis. At the site of a peripheral injury, ribosomes actively translate proteins that serve as signalling molecules that return messages to the soma and act as building blocks for the development of new growth cones [12,13]. Lastly, the activation of intrinsic cellular pathways differs between central and peripheral neurons [14].

Regeneration in the PNS shows that peripheral axotomy commences a program of changes in gene expression with upregulation of a collection of molecules called regeneration-associated genes (RAGs), whereas cutting axons in the CNS results in little to no upregulation of RAGs in those neurons. Injured peripheral neurons orchestrate the RAG response with the help of transcription factors. The first RAG transcription factor (RAG-TF) to be discovered was C-Jun. Later, it was shown that additional TFs, including ATF3, Sox 11, KLF7, and STAT3, can only partially promote axon development [15].

## 3. Barriers to Regeneration

Regeneration and plasticity are strictly monitored by maintaining a balance between growth-promoting and growth-inhibiting factors in the injured mammalian nervous system. Interestingly, both intrinsic and extrinsic factors influence the spontaneous regeneration of neurons. Axonal regeneration at the site of the contusion is severely constrained by the accumulation of growth-inhibiting molecules (semaphorins, netrins, ephrins), deficiency of neurotrophic factors, inflammatory responses, glial scar formation, inhibitory molecules of extracellular matrix (CSPG), and myelin-associated inhibitors (MAIs), such as Neurite outgrowth inhibitor (NOGO), Oligodendrocyte myelin glycoprotein (OMGP), and myelin-associated glycoprotein (MAG). These MAIs bind to the NgR1, Lingo1, Troy, or p75 receptors through the Nogo-66 terminus and are transported to the location of the lesion, where they change the CNS’s capacity for regeneration.

Despite lots of inhibitory molecules, some spontaneous regeneration can occur following spinal cord lesions due to the presence of regenerative molecules. Additionally, fostering remyelination and the creation of synapses close to the location of the lesion may be the most remarkable ways to encourage injured axons to grow over long distances [2,10,16].

## 4. Epigenetic Modifications and Axonal Regeneration

Epigenetic modifiers influence the transcriptional activity of DNA or DNA-associated protein complex by mediating the acetylation/deacetylation of histone protein residues that act as the primary mechanism for axonal regeneration. Acetylation of histone by histone acetyltransferases (HAT) with the help of RAGs allows the opening of the chromatin for the initiation of regeneration. Histone deacetylases (HDACs), on the other hand, contribute to the deacetylation of histone lysine residues, which shuts chromatin expression and decreases RAG expression on chromatin. Modifications in histone acetylation, in particular, help control RAG expression. Histone acetylation at the promoters of specific RAGs rises after peripheral axon contusion but not after central axon laceration. The HAT protein PCAF modulates damage response in peripheral neurons by increasing acetylation. By increasing histone acetylation at specific RAG promoters, overexpression of HAT-PCAF and p300 encourages the healing of spinal cord nerve injuries. TRP53 forms a complex with HATs-PCAF and CBP/p300 in neurons, increasing promoter accessibility to selectively activate multiple RAGs’ expression, including that of *Coro1b*, *Gap-43*, and *Rab13,* and triggering the intrinsic outgrowth program. TRP53, PCAF, or the CBP/p300 complex were shown to be necessary for neurite outgrowth, but TRP53 was only required for axonal regeneration following facial nerve axotomy [17,18,19].

## 5. Molecular Pathways and Therapeutic Targets

Several signalling pathways are attributed to axonal regeneration upon injury or axotomy. Aberration in these pathways contributes to poor axonal growth by disrupting transcriptional machinery. The central regulators that coordinate the regenerative program for intrinsic development are described in detail in this review.

### 5.1. Rapid Injury Signals

#### 5.1.1. cAMP Pathway

Cyclic adenosine monophosphate (cAMP) is a secondary messenger essential for maintaining a neuron’s growth. It is critical for successful axonal regeneration, even in a non-permissive environment. It also controls axon repulsion or attraction in response to environmental cues. It has been seen that in vivo injection of dibutyryl cAMP (db-cAMP) in dorsal root ganglion (DRG) leads to potentiated regeneration of central axons even in the presence of MAIs, thus mimicking conditioning lesions [20]. Another study on zebrafish exhibits an increase in the growth of neurons after induction of exogenous cAMP [21]. Many studies show that cAMP signalling stimulates multiple downstream pathways that drive neurite outgrowth after damage [22].

#### 5.1.2. Calcium/cAMP Pathway

Calcium (Ca^2+^) is the most important factor in regulating cAMP expression following an injury. EG-19 unit (α-subunit) of voltage-gated calcium channels (VGCCs) and internal storage vesicles are two primary sources of the subsequent release of Ca^2+^. After laser axotomy in the *Caenorhabditis elegans* (*C. elegans*) model, Ca^2+^ increases the adenylate cyclase (AC) activity [23,24], triggering cAMP production, which ultimately accelerates the development of growth cones and strengthens the link between damaged axon fragments and the formation of synaptic branches [25].

#### 5.1.3. IL-6/cAMP Pathway

After nerve injury, inflammatory stimulation causes significant axonal regeneration and neuroprotection. Expression of cytokine mediators such as interleukin 6 (IL-6), ciliary neurotrophic factor (CNTF), and leukemia inhibitory factor (LIF) was found to be upregulated in response to injury. Knockout studies have explored the regeneration-promoting effects of these inflammatory molecules. In the adult nervous system, IL-6 is found only in a limited amount in normal situations, but its level rises under injured circumstances and possesses growth-promoting properties. Moreover, the IL-6 knockout study directly indicates the key role of IL-6 in enhancing the regeneration of dorsal column axons in conditioning lesions. cAMP upregulation after injury directly mediates the release of IL-6 and induces activation of downstream pathways for STAT3 and, ultimately *Gap-43* [26,27,28].

#### 5.1.4. PKA/cAMP Pathway

Following injury, cAMP is found to upregulate protein kinase A (PKA) formation, which further promotes the assemblage of cytoskeletal filaments and axonal growth via inhibiting MAG/RhoA-GTPase interaction [4]. On the other hand, it phosphorylates the basic leucine zipper domain (bZip) factor, cAMP response element-binding protein (CREB), which upregulates arginase-1 (*Arg-1*). Upregulation of *Arg-1* enhances polyamine production, which helps to overcome MAG-induced myelin inhibition and stimulates the development of axons [29].

#### 5.1.5. miR-142-3p/AC9/cAMP Pathway

Micro RNAs (miRNAs), non-coding RNAs, regulate gene expression by incorporating them at the 3′ untranslated sites of target mRNA. miR-142-3p plays a key function in regulating the conversion of adenosine triphosphate (ATP) to cAMP by targeting AC9-mRNA (Figure 1). In the sciatic nerve lesion model, miR-142-3p downregulation increased cAMP levels after 9 hrs. As a result, miR-142-3p downregulation promotes AC9 expression, which then raises cAMP levels and improves the regeneration of sensory neurons [30,31].

#### 5.1.6. Pharmacological Agents

Pharmacological treatment approaches can raise cAMP levels in neurons, allowing them to repair more effectively. Intracellular cAMP may be boosted by activating AC using forskolin, which further speeds up the sciatic nerve regeneration. Several neuronal subtypes, including DRG, cerebellar, cortical, and hippocampal neurons, utilize db-cAMP, a non-hydrolyzable cAMP analogue, to overcome MAG and to increase CREB phosphorylation. Brain-derived neurotrophic factor (BDNF) and db-cAMP boost the *Arg-1* expression and antagonize MAG inhibition.

Moreover, repulsive turning reactions in Xenopus (African clawed frog) growth cones are induced by microscopic gradients of the chemo-repellant semaphorin III/D, which may be changed to an attraction on the addition of the cAMP analogue Sp-cAMPS. Surprisingly, Sp-cAMPS therapy boosts calcium levels further, converting MAG-mediated repulsion to attraction to regulate growth cone-turning reactions. The major enzyme responsible for cAMP hydrolysis, phosphodiesterase 4 (PDE4), increases intracellular cAMP levels. Rolipram and roflumilast reduce PDE4 activity to increase cAMP expression [32]. Sorafenib increases sensory conduction function recovery after dorsal column damage by downregulating miR-142-3p [33].

### 5.2. Delayed Injury Signals

#### 5.2.1. DLK-JNK MAPK Pathway

Mammalian zipper protein kinase/dual leucine zipper kinase (DLK) is a mitogen-activated protein kinase kinase kinase (MAPKKK). It plays a central role in neural development, apoptosis, and axonal regeneration in the CNS and PNS [34]. This kinase can further activate cJUN N-terminal kinases (JNK) and p38 MAPK. DLK is a critical component of the sciatic nerve and is required for sensory axon regeneration as well as the activation of regenerative pathways in injured neurons’ cell bodies (Figure 2). The failure of cJUN activation in the DLK knockout mouse model indicates that it is crucial for reorganizing microtubules in the growth cone after axotomy.

A scaffolding protein called JNK-interacting protein 3 (JIP3) connects DLK and JNK-associated axon retrograde trafficking of pSTAT3/JNK toward the injury site [35,36,37]. Hence, DLK is an essential component and target for initiating a pro-regenerative program following injury [38].

#### 5.2.2. Pharmacological Agents

Sarm1, a TIR-domain protein, which functions as a NADase to break down the key metabolite NAD+, is a significant contributor to Wallerian degeneration. The NAD+ biosynthesis enzyme, NMNAT2, hampers Sarm1’s function. MAPK inhibition reduces degeneration triggered by ectopic Sarm1 activation in DRG explants. DLK and NMNAT2 are linked because both are regulated by the PHR ubiquitin ligase. CEP-1347, a pan-mixed lineage kinase (MLK) inhibitor, acts as a key upstream regulator of JNK activation in neurons [39]. Furthermore, palmitoylation regulates DLK binding to JIPs [37], which might affect how they interact with motor proteins. It also facilitates DLK’s interaction with the axonal survival factor NMNAT2 [40].

### 5.3. Transcriptional Factors’ Mediated Pathways

#### 5.3.1. JAK/STAT Pathway

The signal transducer and activator of transcription 3 (STAT3) acts as a ‘jump-start’ for controlling axonal development and remodelling in the PNS and CNS. IL-6, CNTF, LIF, phosphatase and tensin homolog (PTEN), and the suppressor of cytokine signalling 3 (SOCS3) all have a role in STAT3 activation via the Gp130 receptor. It is generally expressed in axons and becomes phosphorylated locally after damage. It then is reassembled in DRG cell bodies, where it promotes axon rejuvenation. Neuropoietic cytokines are produced in response to injury, and Janus kinase 2 (JAK-2) may be activated, which is required for stable STAT3 phosphorylation. DLK deficiency, a key component of the DLK/JNK/MAPK pathway, can prevent p-STAT3 from accumulating in DRG, suggesting that it is required for pSTAT3 translocation [4]. Blockade of PTEN and SOCS3, the small proline-rich protein 1a (*Sprr1a*), and the cell cycle inhibitor *p21/Cip1/Waf1* are therapeutical targets to enhance the expression of STAT3. However, SOCS3 and PTEN blockage are more favorable and common target sites to overcome STAT3 inhibition for axon regeneration [41].

#### 5.3.2. ATF3/CREB Pathway

Normal levels of activating transcription factor 3 (ATF3) expression are low, but PNS laceration increases ATF3 expression in DRG neurons. It has other names, such as LRG-21, TI-241, CRG-5 (in mice), LRF-1 (in rats), and ATF3 (in humans). It belongs to the CREB family, and ATF/CREB can form heterodimers with other bzip proteins such as AP-1 (activator protein-1 transcription factor) family members, including JUN, Fos, and C/EBPs (CCAATenhancer-binding proteins), JUNB, JUND, ATF2, and cJUN. Notably, ATF3 is a stress-stimulated factor and is activated when it binds to ATF/CRE site [42,43]. Different pathways have been found to activate ATF3 expression in response to injury or stress. JNK/SAPK (c-Jun N-terminal kinases/ stress-activated kinase) controls the expression of ATF3 upon peripheral injury and further regulates the expression of different bZip genes to enhance axonal outgrowth [44,45].

After nerve injury, ATF3 activates and targets RAGs, including *cJUN*, *Hsp27*, and *Cap-23,* which is functionally related to *Gap-43*, and *Sprr1a* to induce axonal outgrowth. Similarly, it also mediates the activation of AP-1, which further stimulates *cJUN* activation through JUN N-terminal kinases (JNKs) signalling. c-JUN controls the expression of *galanin* and *CD44*, and *α7β1* integrin to promote axonal regeneration [4]. Thus, ATF3 acts as a point of convergence for different regulators, signals, and transcriptional pathways, suggesting that ATF3 upregulation after an injury is a strong candidate for axon regeneration. Additionally, ATF3 expression correlates with the intrinsic growth capacity of central axons following injury. Sox11 modulates ATF3 expression positively and acts as an upstream target to enhance ATF3 levels and promote axonal growth [46,47].

#### 5.3.3. BMP/SMAD Pathway

The bone morphogenic proteins (BMPs), members of the transforming growth factor (TGF) ligands, are found near the tips of axons and play a role in controlling actin dynamics during dendritic creation and synaptic stability [48,49]. SMADs are involved in neural differentiation, neuronal circuit mapping, and synaptic maturation. Only peripheral branch axotomy (not central branch) increases SMAD1 expression in adult DRG neurons [50,51].

Following peripheral axotomy or contusion, BMP production increases. It was seen that the BMP receptor mediates signalling through SMAD 1 and 4 (Figure 3). Particularly, SMAD 1 plays a chief role in neurite outgrowth. Intra-ganglionic injection of BMP translocates the phosphorylated SMAD to the nucleus and hence modulates its expression. SMAD1/4 recruits other DNA regulatory activators, such as CBP and p300 and activates *Gap-43* to enhance axonal growth [52]. Blockade of BMP in preconditioned or naive DRG neurons inhibits axonal growth, indicating that BMP is necessary for the activation of SMAD. BMP/SMAD1 activation not only increases the growth of dorsal columns in vivo but also promotes intrinsic growth of DRG neurons in in vitro studies using an AAV-based approach. These shreds of data suggest that BMP and SMAD activation work together to promote axonal regeneration. Taken together, BMP is an upstream target of SMAD activation that mediates growth programs in response to injury [52,53,54].

#### 5.3.4. Pharmacological Agents

4-Methylhistamine dihydrochloride (4-MeH), a specific JAK agonist, significantly increases axon growth and remodelling of DRG neurons [55].

### 5.4. Neurotrophins

During development and in the aftermath of injury, neurotrophins, a small family of well-conserved neuropeptides, aid in maintaining the survival of neurons. Peripheral targets such as skin and muscles exhibit baseline expression of neuronal trophic growth factors and are crucial for functional synapse re-establishment following injury. BDNF, NGF, and NT3/4/5 are the most well-known members of this family. NGF is the most researched of all neurotrophins [56,57]. Tyrosine kinase receptors (TrkA, TrkB, TrkC) and low-affinity p75^NTR^ receptors interact with members of this family. Phospholipase C-γ (PLC-γ), phosphatidylinositol-3-kinase (PI3K), and Ras pathways are activated when neurotrophins bind to Trk receptors to mediate gene expression (Figure 4) [58].

#### 5.4.1. Ras/ERK Pathway

Nerve growth factor (NGF), the first-ever reported neurotrophin, helps maintain phenotypes, promotes PNS development, and ensures the functional unity of cholinergic neurons in the CNS (for detailed review, see [59]). It binds to the Trk receptor (Tropomyosin receptor kinase) and activates Ras/ERK signalling. Activated Ras stimulates Raf (serine/threonine kinase), MEK (MAP/ERK kinase), and, lastly, ERK (extracellular signal-regulated kinase). ERK belongs to the MAPK (mitogen-associated protein kinase) family and plays an important role in neurite outgrowth. Robust and rapid activation of ERK following nerve injury in Schwann cells (SCs) has been monitored at the injury site. A higher level of ERK has been seen in both infectious and inherited peripheral neuropathies models. It also plays a chief role in neurological healing and remyelination of neurons. Ras/ERK signalling enhances axon growth and activates anti-apoptotic protein Bcl-2 via CREB to promote axonal survival [8,60,61]. Hence, the upregulation of ERK is the leading candidate for the Ras/ERK pathway [62].

#### 5.4.2. PI3K/AKT Pathway

The PI3K/AKT pathway is an important survival mechanism that controls signalling, differentiation, and axonal growth. The interaction of NGF and Trk receptors activates PI3K via the Ras and Gab-1 adaptor proteins, allowing PI3K phosphorylation and activation of downstream molecules such as AKT and ERK1/2.

Activated AKT reduces Fas ligand (FasL) expression by suppressing the pro-apoptotic transcription factor Bad (inhibitor of Bcl-2 anti-apoptotic protein), and Forkhead. Phosphorylated AKT, on the other side, promotes the assemblage of cytoskeletal filaments and axonal outgrowth [63,64]. It was seen in PC12 rat pheochromocytoma cells that PI3K activates Rac1, which translocates RhoA from the nucleus to the cytoplasm, forming a complex with GDP dissociation inhibitors (GDIs), rendering it unable to halt axon renewal [65]. Knockout studies of PTEN demonstrate accelerated outgrowth in both in vitro and in vivo models of sensory neurons. The inhibition of PTEN acts as a therapeutic target to increase neurite growth, suggesting that upregulation of AKT is required for healing damaged nerve responses [66,67,68].

#### 5.4.3. Dock/BDNF Pathway

Rho-GTPase family members (Rac-1, RhoA, and Cdc42) regulate the actin cytoskeletal organization. All members of this family are inactivated in GDP-bound form while activated in the form of GTP. GTPase activating proteins (GAP) and guanine nucleotide exchange factors (GEFs) regulate Rho activation and inactivation by converting it into a GDP-bound form and stimulating nucleotide release, respectively [69].

The Rho-GEF family (Rho-guanine nucleotide exchange factors) includes Dock1 (Dock180)-related proteins, although Dock 1 has no PH/DH (pleckstrin homology/Dbl-homology) domains but exhibits the DHR-1/DHR-2 evolutionary proteins region. Dock (1–4) has conserved several amino acid regions in their DHR-2 domains that are essential for GEF activity. Moreover, in vivo and in vitro studies have proved Dock-3 as a critical component in neuron maintenance. It participates in axonal growth via downstream BDNF-TrkB signalling and is essential in CNS developmental processes such as synaptic pruning [70].

Furthermore, Dock-3 forms a complex with the WAVE-1 and Fyn-SH3 domain at the cell membrane level. In this notion, it is clear that Dock-3 expression is directly linked to WAVE-1 expression to promote the regeneration of injured optic nerves. BDNF expression is linked to Rac-1 activation, which in turn fosters neurite development. Dock-3 also promotes WAVE protein recruitment and hinders the long-term activation of WAVE/Rac1 for actin polymerization (Figure 5). Taken together, Dock upregulation has been shown to cure various glaucoma conditions. Interestingly, BDNF and Dock-3 have a synergistic relationship upstream of Rac-1 and WAVE proteins, indicating BDNF as a possible therapeutic target for the upregulation of Dock-3 and their associated mediators to enhance axonal outgrowth [71,72,73]. A list of intrinsic regulators is shown in Table 1.

#### 5.4.4. Pharmacological Agents

At nanomolar doses, bisperoxovanadium compounds (bpVs) selectively inhibit PTEN [74,75]. Interferon-β enhances NGF protein; thus, it can initiate NGF/Trk signalling in damaged axons to ameliorate functional recovery [76,77].

BRD4, a BET bromodomain reader of acetyl-lysine histones, is an essential component that enhances BDNF synthesis and memory after HDAC inhibition. HDAC2 and HDAC3 knockdown increased BDNF-mRNA expression but not the expression of other HDACs, whereas BRD4 knockdown blocked these effects. The BDNF promoter-specific targeting of BRD4 by the dCas9-BRD4 locus enhanced BDNF-mRNA. RGFP966, a pharmacological inhibitor of HDAC3, was suppressed by JQ1, which increased BDNF expression and BRD4 binding to the BDNF promoter (an inhibitor of BRD4). Following RGFP966 treatment, H4K5ac and H4K8ac alterations, as well as H4K5ac enrichment at the BDNF promoter, were enhanced according to documented epigenetic targets of BRD4 and HDAC3 [78].

**Table 1 biomedicines-10-03186-t001:** Intrinsic regulators for axonal regeneration under injury condition.

Regulator	Studied in	Axonal Regeneration	Studied by
**Regeneration-associated transcription factors**
**STAT3**	Dorsal root ganglion (DRG) neurons	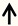	[79]
**cJUN**	DRG neurons	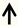	[80,81,82]
**KLF4**	Retinal ganglion cells (RGC) neurons	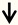	[4,83]
**KLF6**	RGC neurons	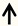	[11,84,85]
**KLF7**	Corticospinal tract (CST) neurons	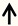	[11,84,85]
**KLF9**	RGC neurons	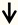	[86]
**SMAD1**	DRG neurons	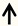	[87]
**ATF3**	DRG neurons	* 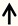	[46,88]
**CREB**	DRG neurons	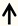	[89]
**AP-1**	DRG neurons	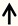	[90]
**Intrinsic growth-mediating molecules**
**cAMP**	DRG neurons	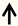	[9,22,28,91]
**mTOR**	CST neuronsRGC neuronsDRG neurons	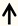	[92,93,94,95]
**PTEN**	CST neuronsRGC neurons	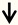	[96,97]
**SOCS3**	RGC neurons	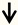	[98,99]
**TSC1 (hamartin)**	RGC neurons	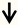	[100,101]
**TSC2 (tuberin)**	RGC neurons	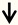	[102,103]
**GSK-3β**	DRG neurons	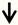	[104]
**LIF**	Sensory neurons **	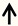	[15]
**IL6**	Sensory neurons,Schwann cells	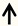	[105,106]
**CNTF**	Distal nerve stump of DRG neurons	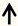	[106,107,108]
**IGF-1**	RGC neurons	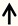	[109]
**PI3K**	DRG neurons	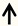	[110]
**ERK**	DRG neurons	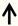	[111,112,113]
**FAK**	RGC neurons	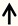	[114]
**DLK**	DRG neurons	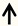	[115,116]
**Regeneration-associated genes (RAGs)**
** *Galanin* **	DRG neurons	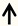	[117,118]
** *Integrin* **	DRG neurons	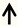	[119]
** *Gap-43* **	DRG neurons	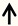	[19,120]
** *Cap23* **	DRG neurons	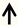	[121]
** *Hsp27* **	DRG neurons	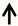	[121,122,123]
** *P21/cip1/Waf1* **	DRG neurons	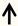	[41]
** *Sprr1a* **	DRG neurons	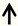	[4]
** *Crmp2* **	Motor neurons	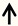	[124]
** *Map1b* **	Growth cone	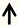	[125]
** *Spry2* **	Sensory axons	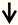	[126,127]
** *Pdcd4* **	Spinal cord neurons	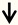	[128]
** *Dickkopf1* **	DRG neurons	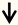	[129]
**Rho GTPases**
**Rho-A**	DRG neurons	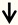	[130]
**Rac1**	RGC neurons	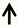	[131]
**ROCK**	RGC neurons	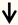	[132]
**miRNAs**
**miR-142-3p**	DRG neurons	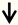	[133]
**miR-21**	DRG neurons	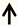	[134]
**miRNA-431**	DRG neurons	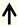	[135]
**CNS inhibitors**
**Nogo**	RGC neurons	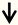	[136]
**MAG**	Central nervous system (CNS) neurons	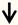	[137]
**Omgp**	CNS neurons	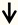	[137]
**CSPGs**	CNS neurons	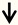	[15]
**Semaphorins**	CNS neurons	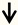	[138]

* In PNS lesion, not in CNS lesion. ** Schwann cells upregulate it after injury even though it is not typically present in the nervous system.

### 5.5. AKT/mTORC1/p70S6K Pathway

The mammalian target of rapamycin (mTOR) is a 268 kDa protein and serine-threonine kinase that controls cell growth and regeneration via balancing nutrient availability. It is also known as RAFT, RAPT, and FRAP. When the PI3K/AKT pathway activates mTORC1, it stimulates the expression of p70S6K, which limits muscle atrophy and promotes axonal regeneration. Through the AKT/mTORC1/p70S6K pathway, mTOR can influence the regeneration potential of growth cones in adult corticospinal neurons (Figure 6) (for details, please see [139,140]).

It was seen that mTOR increases axon regeneration in the PNS through the deletion of negative regulators, including PTEN, hamartin (TSC1), and tuberin (TSC2) [4,57]. Interestingly, TSC2 inhibits axon growth by inactivating Rheb (Ras homolog enriched in the brain), which is a GTP-binding protein. Typically, activated Rheb acts as a direct activator of AKT-mediated mTORC1 signalling as mentioned by in vivo study of the ischemic spinal cord injury model [141]. Indeed, the balance of inhibitors and activators determines the pace of growth; it is necessary to target these inhibitors in order to offer the optimum regenerative environment for damaged neurons. PTEN knockdown or long-term inhibition, on the other hand, has been linked to neoplasia, including malignant glial tumors and hamartomas, according to prior research [63,100,142]. TSC-1 and TSC-2, which stimulates axonal growth, can be inhibited by activated AKT. Conclusively, these shreds of evidence suggested that the upregulation of AKT, as opposed to PTEN or TSC1 and TSC-2, is a more useful and appealing therapeutic target due to fewer adverse effects [139]. It is suggested that there is an AKT-independent route for axonal regeneration because PTEN deletion results in considerably more potent axonal regeneration, whether combined with AKT overexpression or GSK3β deletion [143]. Blockage of *Pdcd4* has also been shown to be an important technique in preventing muscle atrophy by boosting elF4B-induced mRNA and thus assisting axonal regeneration [144].

#### Pharmacological Agents

Sirolimus and everolimus, two mTOR inhibitors, have the potential to deliver tailored treatment. In the United States and Europe, Everolimus is the first mTOR inhibitor to be licensed as a therapy option [145].

### 5.6. IGF-1/GH Pathway

Hypothalamic growth hormone-releasing hormones (GHRHs) act on the pituitary gland to maintain the release of growth hormone (GH). These hormones have therapeutic potential due to their role in accelerating axonal regeneration and maintaining denervated muscle and SCs before reinnervation. It works by releasing insulin-like growth factor-1 (IGF-1) into the bloodstream, which is synthesized mainly in the liver and also in peripheral tissues (for a detailed study, see [146,147]). IGF-1 is involved in the regeneration and survival of CNS neurons and the stimulation of neurite outgrowth [148,149]. In addition, IGF-1 mediates axon elongation and is also involved in developing neurons’ survival, proliferation, and synaptogenesis [150].

After nerve trauma, IGF-1 production is upregulated locally, which in turn upregulates the expression of *Gap-43* and focal adhesion molecules via the PI3K pathway. Taken together with these changes, IGF-1 promotes motility of the neurite growth cone and prevents neuronal apoptosis. Overexpressed IGF-1 also upregulates the mTOR/AKT pathway, MuRF1, and MADbx to counteract the denervation-induced muscle atrophy [151,152]. IGF-1’s supportive role in peripheral nerve injury (PNI) recovery in animal models is strongly supported by research. IGF-1 can slow the pace of muscle atrophy, promote axonal development, and reduce SC apoptosis [153].

IGF binding proteins (IGFBPs) facilitate the binding of IGF molecules with the IGF-1R (tyrosine kinase receptor) and function by activating numerous pathways such as PI3K/AKT, mTOR, and MEK/ERK (Figure 7). The interplay of IGF/IGFBP/IGF-1R causes calcium influx and PI3K phosphorylation, as well as mTOR activation to promote axon growth. Additionally, receptor activation causes phosphorylation of receptor tyrosine residues, which activates Ras GTPases and their molecules to activate Raf, MEK, and eventually ERK. Rho-GTPase, Cdc42, and glycogen synthase kinase 3 beta (GSK-3β) are downstream targets of IGF-1 and play a role in stabilising microtubule polymerization. Similarly, GH exerts its action by activating the JAK/STAT pathway to enhance neuronal survival and outgrowth [147,154]. In a mouse model of nerve damage, GH reduces the negative consequences of muscle denervation and improves not just functional recovery but also motor axon reinnervation. These findings suggested that increasing GH levels might be a potential therapy option for improving motor function recovery [155].

### 5.7. GSK3β–CLASP/APC Pathway

GSK-3β, a serine-threonine kinase, can phosphorylate several proteins, including microtubule-associated proteins (tau and MAP1B), collapsin response mediator protein 2 (CRMP2), and cytoplasmic linker-associated protein 2 (CLASP2), when activated. GSK3β activity can be justified by mediating their phosphorylating sites [156]. It is normally phosphorylated at its Tyr216 residues of the kinase domain to become active when it is at rest, whereas phosphorylation of Ser9 residues is responsible for the inactivity of GSK3β [157,158]. Integrin-mediated inhibition of GSK3β can stimulate the CLASP/APC pathway where CLASP2 (a cytoplasmic linker protein) plays a role in axonal microtubule stability for potential axonal regeneration [104,159]. CRMP2, another substrate of GSK3β, promotes microtubule organization and enhances axon elongation by binding to tubulin heterodimers. This binding is reduced upon phosphorylation of GSK3β. GSK3β can stimulate CRMP2 phosphorylation at the Thr514 residue, while ROCK phosphorylates CRMP2 at the Thr555 residue. ROCK-mediated CRMP2 phosphorylation can downstream Nogo-66 and MAG thereby counteracting myelin inhibition. So, CRMP2 is a central therapeutic target for reversing the inhibitory effect of myelin inhibitors for axon regeneration [159]. In addition, protein phosphatase 2A dephosphorylates CRMP2 and promotes axon outgrowth [160], whereas CRMP2 is active only when GSK3β is inactive. Furthermore, activated GSK3β can mediate the expression of MAP1B to regulate the growth of cytoskeletal filaments [91,158]. The pCRMP2 improves axonal regeneration after GSK3β knockout following optic nerve damage. GSK3β mediates axonal regeneration by targeting specific substrates, suggesting that it might be a therapeutic target to improve regeneration [161].

#### 5.7.1. GSK-3β/Wnt Pathway

Wnt ligands, also known as glycoproteins, have an important and diversified function in cell growth, development, and various human disorders [162]. Wnt-mRNA expression is not detectable in the intact spinal cord, whereas Wnt 1, 4, and 5a immediately surround the lesion site after spinal cord contusion. These molecules also increase Wnt-Ryk signalling, which inhibits motor neuron renewal [163]. Wnt ligands target particular receptors to execute their activities; for example, Wnt-4/Ryk ligand interaction is found to inhibit sensory neuron regeneration, whereas Wnt-LRP5/6 interaction is found to promote neurite outgrowth [164]. Consistent with this interaction, downregulation of Wnt-Ryk signalling is an attractive therapeutical approach to promote the plasticity of cortico-spinal motor neurons and robust axonal growth following injury [163,165,166].

Wnt ligands can block GSK-3β activity and promote re-myelination of facial and sciatic nerves through Wnt/β-catenin signalling. In addition, Wnt-GSK3β interaction has been shown to enhance axonal regeneration as much as neurotrophins [161,167]. GSK-3β inhibition can stimulate the migration of SCs, MAG, myelin-related genes, cyclin-D1, nicotinic-acetylcholine receptors, and muscle gene myogenin to promote axon regeneration by reducing muscle degeneration. This suggests that blocking GSK3β or using inhibitors to promote myelination and nerve regeneration is a valuable option [168]. Moreover, earlier research has shown that upregulation of the Wnt/β-catenin pathway improves axonal regeneration after damage [139,169,170,171,172].

#### 5.7.2. miRNA-431/Kremen/Wnt Pathway

Kremen (1/2), a transmembrane protein, has a kringle domain and acts as a receptor for Dickkopf1 (Dkk1) [129]. Dkk1 and Kremen-1 form a complex that further inhibits Wnt signalling (Figure 8. By blocking Kremen 1, overexpressed miR-431 can significantly suppress the expression of other genes, such as *Braf* and *Zkscan 1*, thus promoting axonal growth [129,135].

## 6. Other Injury Signals

### 6.1. Regeneration by Astrocytes/Inflammatory Mediators

In mammals, the inflammatory response occurs instantaneously after a CNS contusion and plays an important role in providing restorative output. Microglia are the inflammatory mediators of the CNS and present the first-line defensive approach against trauma. After activation, they undergo morphological modifications such as ramification into amoeboids. They begin to proliferate and migrate toward the injury site, thereby initiating the production of pro-inflammatory and anti-inflammatory markers such as cytokines [173,174]. Moreover, macrophages and neutrophils are recruited to the injured area from the periphery. Together with microglia/macrophages, reactive astrocytes also participate in the organization of regenerating-inhibiting glial damage [91,175,176].

Upon optic nerve contusion, microglial cells of the retina reactivate themselves and stimulate the release of blood-borne macrophages and neutrophils, which secrete inflammatory mediators such as oncomodulin (Ocm). Conversely, inflammatory stimulation provokes reactive astrocytes to promote axon regeneration and neuroprotection via mTOR and JAK/STAT3 signalling pathways. SOCS3 acts as an inhibitor of the JAK/STAT3 pathway, whereas PTEN is the inhibitor of the mTOR pathway. Deletion of SOCS3 and PTEN may prove a valuable therapeutic target to enhance axonal regeneration. Notably, SOCS3 expression can be opposed by cAMP level; hence, an elevated level of cAMP can enhance neurite regeneration (Figure 9). Additionally, overexpression of Ras homolog regulator (Rhen1), which is enriched in the brain, can boost mTOR activity and, thereby, RGC axonal regeneration [176].

### 6.2. miRNA-21 and Axon Regeneration

Micro RNAs (miRNAs) are encoded endogenously to regulate gene expression by inhibiting or increasing the degradation of the translational protein. Due to their role in neuronal development, regeneration, and degeneration, miRNAs have been explored as a therapeutic tool for various diseases [177,178,179,180,181]. miRNA-21 (miR-21) can enhance the intrinsic growth capacity of damaged neurons after nerve denervation [134]. PTEN, *Pdcd4, Spry2*, and *Tpm1* are the targets of miR-21, through which it mediates neurite outgrowth [182]. Different studies revealed that miR-21 regulates *Spry2* in cardiomyocytes [183] and PTEN in hepatocellular carcinoma [184]. TGF-β1 mediates miR-21 proliferative and apoptotic activity by targeting the PI3K/AKT/mTOR pathway and promotes astrocytes activation after spinal cord injury (SCI) both in vitro and in vivo. Moreover, TGF-β1 activates SMAD2/3 and upregulates miR-21 to prevent astrocytic scar formation, which is the leading cause of hindrance of axonal regeneration (Figure 10). Herein, the upregulation of miR-21 and downregulation of PTEN are promising therapeutic targets to regulate neural regeneration [182]. Moreover, it was seen that miR-21 promotes axonal growth and maintains the number of SCs by interacting with TIMP3, EPHA4, and TGFβ1 during nerve repair [185].

### 6.3. Integrin/FAK Pathway

The integrin receptor family is vital for developing tissues, nervous system formation, immune responses, and regeneration of axons. There are two subunits in integrin receptors: alpha and beta. In mammals, a total of eighteen α and 8 β-subunits have been identified [186,187,188].

Activated integrins specifically bind to particular extracellular matrix ligands (ECM) and stimulate signalling to regulate the actin cytoskeleton [189]. Tenascin-C is a ligand for integrins present in the CNS that can be regulated by reactive astrocytes. Tenascin-C interacts with an α9β1-integrin receptor in neurons to achieve axon regeneration in CNS (for details, see [190,191]). It was found that α9 delivery (viral vector-mediated) in DRG causes the localization of integrins in axons; therefore, it induces neurite growth [119,192,193,194].

Once integrin is activated, it interacts with tenascin C and stimulates FAK, which further activates downstream molecules, including PI3K, AKT, Src kinase, and RhoA. The lesion site is also enriched with molecules such as NogoA, CSPGs, MAG, and semaphorins class III (Sema3s). These molecules inhibit integrin signalling, and block phosphorylation of FAK to prevent axon regeneration via binding to their respective receptors to mediate axonal guidance and growth cone dynamics (Figure 11). An appropriate elevated integrin expression can serve as a therapeutic approach for CNS axonal regeneration. To overcome integrin inactivation, PTEN and GSK-3β blockage are considered as the target sites to elevate integrin enrichment at the injury site in promoting axonal growth [195,196,197].

It has been noted that in *C. elegans*, the RhoA GTPase-ROCK pathway stimulates MLC-4 phosphorylation to enhance axon regeneration. Axonal injury activates TLN-1/talin via the cAMP-Epac-Rap GTPase cascade, which results in integrin inside-out activation and promotes axonal regeneration by activating the RhoA signalling pathway. Src-1 activates EPHX-1/ephexin RhoGEF in this pathway after integrin activation by phosphorylating the Tyr-568 residue in the autoinhibitory domain. These findings show that the Src-RhoGEF-RhoA axis controls axon regeneration by following an integrin signalling network [198].

### 6.4. RhoA/ROCK/LIMK Pathway

The RhoA/Rho kinase pathway has a role in the regulation of inflammatory responses and the production of cytokines, including interleukin 2 (IL-2), interleukin 1 beta (IL-1β), tumor necrosis factor-α (TNF-α), and CXC chemokines [199]. Rho-GTPases allow the interaction of extracellular ligands with the growth cone and modulate neurite outgrowth [200]. RhoA-GTPases and their downstream effectors, including ROCK, along with MAGs and chondroitin sulfate, activate PNI [201,202]. ROCK negatively affects the regeneration of axons by modulating the growth cone dynamics [132]. ROCK1 and ROCK2 are two isoforms of ROCK. ROCK1 mostly exists in non-neuronal tissues, while ROCK2 is found in the spinal cord and brain [203,204].

Guanine nucleotide exchange factors (GEFs), GDP dissociation inhibitors (GDIs), and GTPase activating proteins (GAPs) play a crucial role in regulating the phosphorylation of Rho-GDP into Rho-GTP. ROCK also inactivates MLCP (myosin light chain phosphatase), which dephosphorylates the myosin light chain. ROCK-induced MLCP inactivation enhances myosin’s phosphorylation that causes growth cone collapse and axon growth inhibition [205]. LIM domain kinase (LIMK) is considered a downstream target of ROCK2. LIMK activation results in a collapse in growth cone stability. This effect is achieved after cofilin phosphorylation. Upon phosphorylation, cofilin becomes inactivated and further incapable of organizing actin filaments. Nogo/Lingo-1/p75^NTR^ or ephrin/semaphorin receptors are responsible for the activation of ROCK by binding with myelin-based inhibitors such as NOGO, MAG, OMGP, semaphorins, and ephrins. ROCK activation not only plays a role in the regulation of axon guidance but is also important in promoting regenerative machinery in mammalian CNS. Apart from this, ROCK has deleterious effects on cell survival by activating Fas or ROCK homolog PTEN (Figure 12). ROCK inhibition looks to be a promising treatment option for a variety of neurodegenerative illnesses in this regard. CRMP2 has also been discovered to increase microtubule assembly, which is important for axon polarity and outgrowth. ROCK2 can phosphorylate and activate CRMP2, hence having growth-promoting action by regulating growth cone morphology [203,206,207].

#### Pharmacological Agents

In an optic nerve damage model, blocking the RhoA/ROCK pathway with Y-27362 dramatically enhances axonal sprouting and locomotor functional recovery [208]. An experiment examined the effects of Y-27632 on peripheral motor and sensory neurons that had developed in the presence of CSPGs that hindered axonal extension. In vitro, motor neurons showed a more favorable response to Y-27632 than that of sensory neurons in a non-growth-permissive environment. These variations were related to different RhoA expressions and activation patterns in sensory and motor axons. After systemic treatment of high doses of Y-27632, the regeneration of motor axons was significantly enhanced in vivo, in contrast to the regeneration of sensory fibers. The findings demonstrate that the RhoA/ROCK pathway differentially influences both sensory and motor axon regeneration, with motor neurons responding more quickly than sensory neurons under a growth-inhibitory environment. 

In addition, the regeneration of axons was investigated in symptomatic and asymptomatic SOD1G93A animals after the sciatic nerve was crushed. Axonal regeneration was severely hampered when symptomatic SOD1G93A animals were compared to presymptomatic SOD1G93A mice and wild types. Treatment with Y-27632 enhanced motor axon functional and morphological measurements following the sciatic crush in all examined situations. Although Y-27632 therapy enhanced neuromuscular junction axonal reinnervation, it did not increase the lifespan of symptomatic SOD1G93A animals [209]. To alleviate neurofunctional impairments, bFGF might inhibit RhoA by activating Ras-related C3-botulinum toxin substrate 1 (Rac1). Thus, inhibiting RhoA/ROCK activation in case ofneurological illnesses is critical for neuroprotection and neurogenesis [210].

### 6.5. POSTN/Integrin Pathway and Axon Regeneration

Periostin (POSTN), an extracellular matrix protein, belongs to the fasciclin family [211]. It has been linked to the regeneration of astrocytes. During the early stages of development, it is expressed in various tissues, including the CNS. Its expression rises in adult tissues after acute injury, including shattered bone infarcted myocardial tissues, and during wound healing [212,213]. A recent study shows that POSTN stimulates neurite outgrowth by activating AKT and FAK [214].

Similarly, POSTN is involved in cell proliferation, survival, migration, and adhesion by interacting with integrins α5/β3 and α5/β1, and it also promotes downstream activation of PI3K, FAK, and AKT [211,215]. Moreover, POSTN can bind to other extracellular matrix proteins such as tenascin C, fibronectin, and collagen 1 [216]. Through the FAK/AKT signalling pathway, the POSTN/integrin axis is activated in the injured spinal cord and promotes macrophage migration toward the injury site. It inhibits pericyte proliferation to minimize scar formation, which is thought to be the principal hindrance to CNS axonal regeneration [217]. The ability of POSTN to induce neuronal development is further enhanced when the FAK pathway is activated (Figure 13). As a result, upregulating FAK/AKT is a therapeutic strategy for preventing pericyte proliferation [205,218].

## 7. Conclusions

Both central and peripheral axons are very delicate and highly sensitive structures, and even mild stress or pressure can injure them. Axonal injury can be caused by neurotoxins, traffic accidents, acute lacerations, and neurological disorders resulting in the loss of functions. Failure of axonal regeneration can lead to lifelong disability. Peripheral nerves can regenerate in the presence of a supportive environment provided by SCs. Another important factor is the fast clearance of cellular debris in the distal part through the involvement of macrophages that facilitate the regeneration. On the other hand, in case of injury to CNS, oligodendrocytes fail to provide the supportive environment required for regeneration; instead, they secrete inhibitory molecules. Central nerves cannot regenerate since they do not have enough intrinsic growth factors. On the other hand, excessive stress on peripheral nerves may irreversibly damage axons and inhibit axon regrowth. After an injury, several signalling molecules are activated and travel toward the damaged area, potentially speeding up the regeneration process. Among them, signalling pathways such as cAMP, JAK/STAT, BDNF, ERK, PI3K/AKT, mTORC1, integrin/FAK, and Rho/ROCK are common and are thought to initiate the process of regeneration. Furthermore, astrocytes, inflammatory cytokines, and microRNAs all have a role in growth cone stability. Upregulation of cAMP, STAT3, AKT, BDNF, integrin, miR-21, miR-431, and downregulation or blocking of GSK-3β, PTEN, SOCS3, TSC1, TSC2, miR-142-3p, *Pdcd4*, *Spry2*, *Tpm1*, and myelin-associated inhibitors can alter the fate of regenerating axon and therefore are considered as potential therapeutic targets to restore the function of injured axons. However, when it comes to the execution of therapeutic approaches for functional axon recovery, a valid and affordable clinical approach is still nonexistent.

## Figures and Tables

**Figure 1 biomedicines-10-03186-f001:**
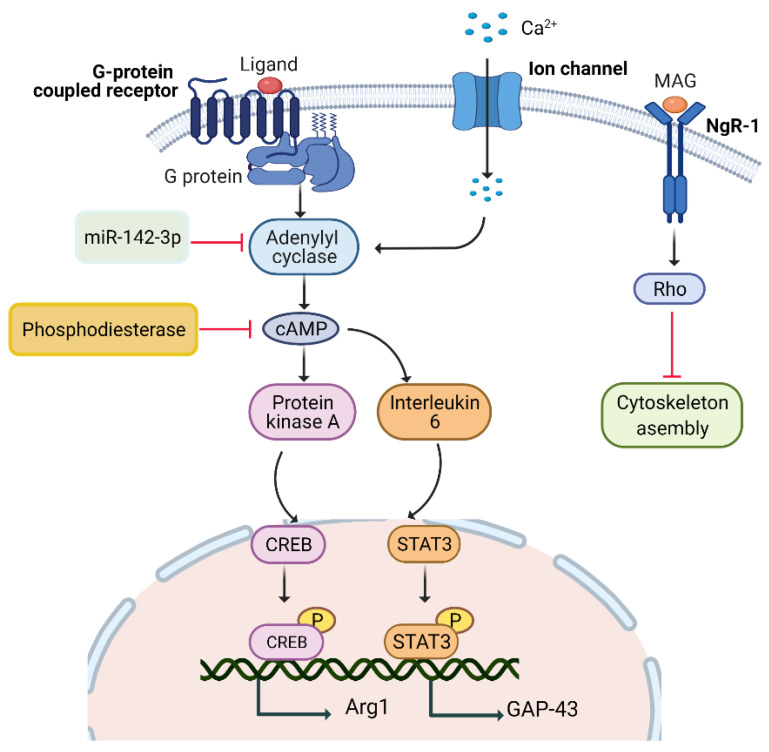
cAMP pathway. Peripheral nerve injury triggers calcium release that upregulates cAMP level via enhancing AC9 expression and regulates three pathways in DRG neurons to promote neurite outgrowth. miR-142-3p, PKA, Rho and phosphodiesterase are the target sites of AC9, Rho, cytoskeletal assembly, and cAMP, respectively, and can be used as therapeutic targets to enhance neurite outgrowth. cAMP: cycline adenosine monophosphate; AC9: adenylyl cyclase 9; DRG: dorsal root ganglion; PKA: protein kinase A; MAG: myelin-associated glycoproteins; NgR1: NOGO receptor; CREB: cAMP response element-binding protein; STAT3: signal transducer and activator of transcription 3; *Gap-43*: growth-associated proteins; *Arg-1*: arginase-1.

**Figure 2 biomedicines-10-03186-f002:**
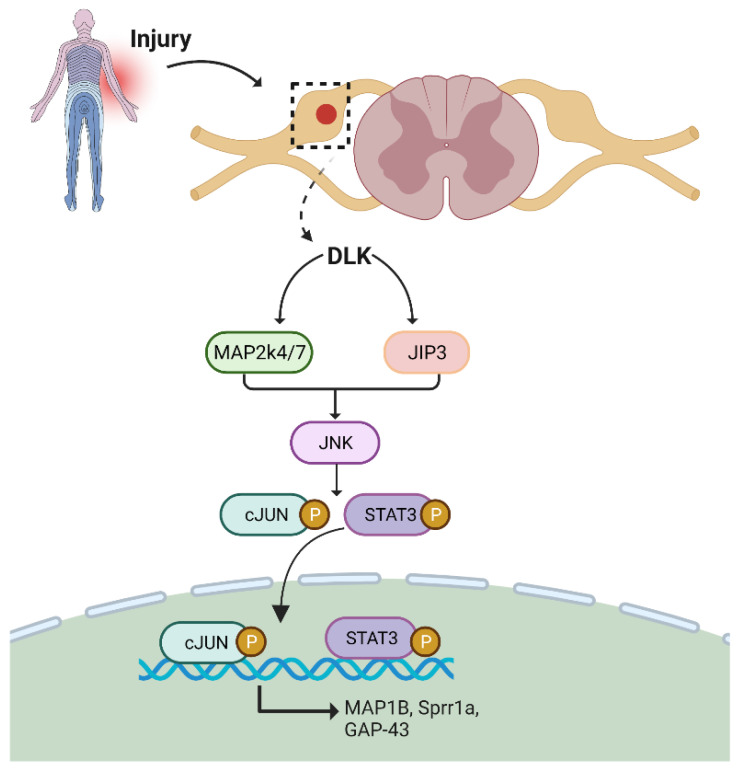
DLK/JNK/MAPK pathway. DLK acts on JIP3 and MAP2k4/7 to activate JNK-mediated STAT3 and cJUN to enhance axon regeneration. DLK: dual leucine zipper kinase; JIP3: JNK-interacting protein 3; MAP2k4/7: mitogen-activated protein kinase; JNK: c-Jun N terminal kinase; STAT3: signal transducer and activator of transcription 3; *Map1b*: microtubule-associated proteins; *Sprr1a*: small proline-rich protein 1A; *Gap-43*: growth-associated proteins.

**Figure 3 biomedicines-10-03186-f003:**
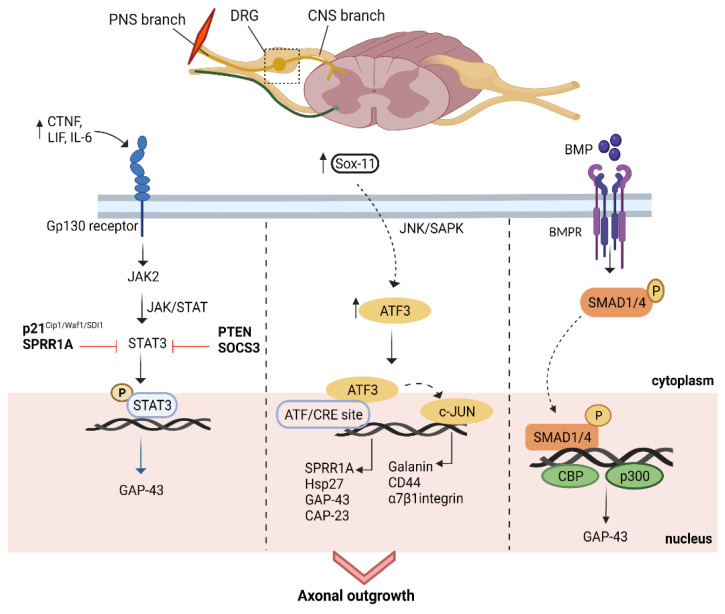
Transcriptional factors’ mediated pathways. Following injury, different intrinsic mechanisms are stimulated and mediated by regeneration associated transcriptional factors. First, injury upregulates the expression of IL-6, LIF, and CTNF in DRG neurons that interfere with Gp130 receptors to activate the JAK/STAT pathway to promote axon growth. Similarly, ATF3 increases upon injury and undergoes JNK signaling to activate AP-1 and c-JUN-mediated RAGs to increase axon outgrowth, respectively. Moreover, phosphorylation of SMAD 1 and 4 causes p300 and CBP recruitment and enhances *Gap-43* expression to promote axon regeneration. IL-6: interleukin 6; LIF: leukemia inhibitory factor; CTNF: ciliary neurotrophic factor; DRG: dorsal root ganglion; ATF3: activating transcription factor 3; JNK; AP-1: activator protein-1; *Gap-43*: growth-associated protein; SMAD: protein; p300/CBP: co-activating proteins; BMP: bone morphogenic proteins; BMPR: bone morphogenic proteins receptors; *Sprr1a/Hsp27/Gap-43/Cap-23/Galanin/α7βintegrin*: regeneration-associated genes.

**Figure 4 biomedicines-10-03186-f004:**
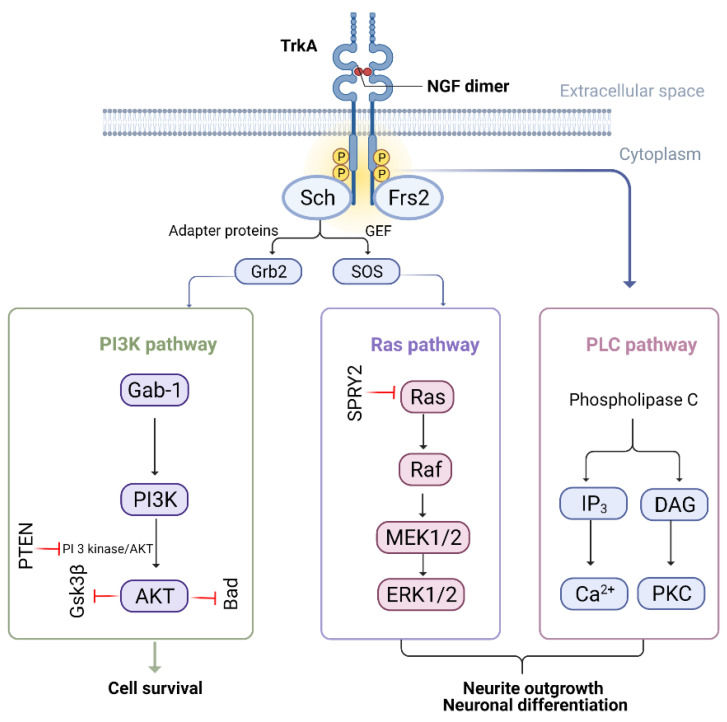
NGF-mediated pathways. Following NGF binding to tropomyosin, receptor kinase activates Ras/ERK and PI3K/AKT, which subsequently leads to increase neuronal survival and, ultimately, axonal growth. TrkA: tyrosine kinase receptor; NGF: nerve growth factor; GEF: guanine nucleotide exchange factor; SOS/Grb2: adaptor proteins; Frs2: fibroblast growth factor receptor substrate 2; ERK: extracellular signal-regulated kinase; PI3K: phosphatidylinositol-3-kinase; AKT: protein kinase B; DAG: diacyl glycerol; PKC: protein kinase C; IP3: inositol triphosphate; Ras/Raf/MEK: protein kinases; Gab-1: GRB2-associated-binding protein 1.

**Figure 5 biomedicines-10-03186-f005:**
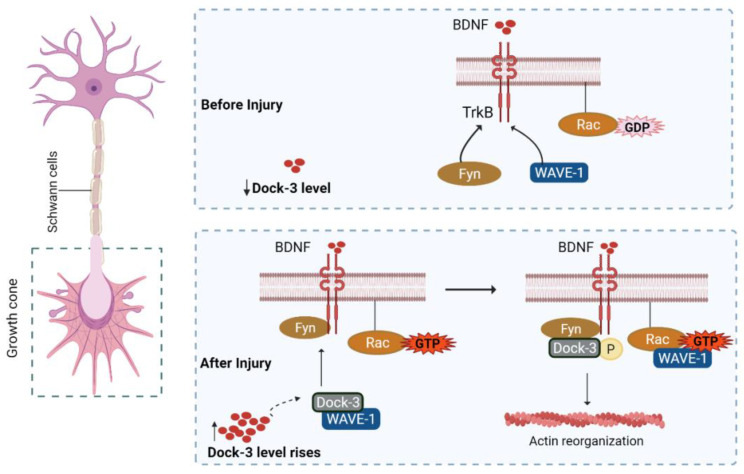
Dock-mediated BDNF signaling. Upon injury, Dock-3 expression increases at the injury site that recruits WAVE-1 to form a complex with Fyn. This complex interacts with Trk-B receptors to modulate Rac-GDP to Rac-GTP and mediates actin organization for axonal growth. BDNF: brain-derived neurotrophic factor; TrkB: tyrosine kinase receptor; GDP: guanosine diphosphate; GTP: guanosine triphosphate; Dock3/Rac/Fyn/WAVE-1: protein.

**Figure 6 biomedicines-10-03186-f006:**
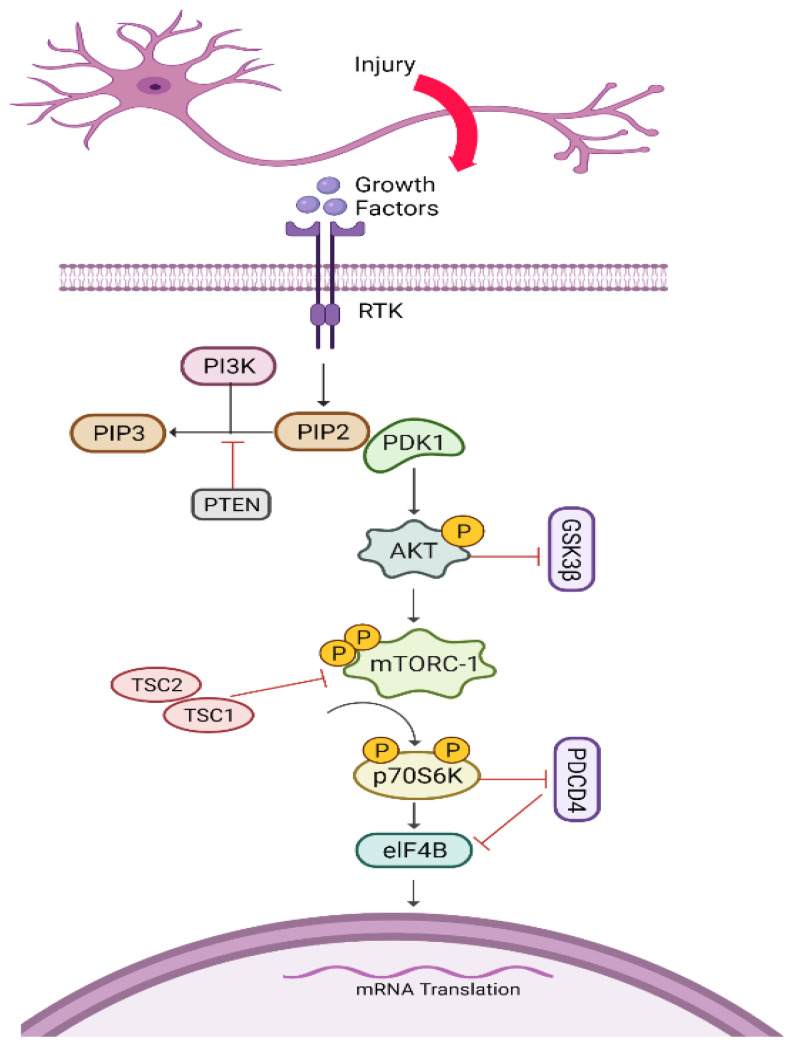
AKT/mTORC1/p70S6K pathway. After nerve injury, several growth factors participate to initiate a cascade of reactions for axonal growth. They bind to receptor tyrosine kinase and activate PI3K/AKT-mediated mTORC1, which further phosphorylates S6. pS6 stimulates elG4B to prevent muscle atrophy and thus increases axon regeneration. RTK: receptor tyrosine kinase; PI3K: phosphatidylinositol-3-kinase; PIP3: phosphatidylinositol (3,4,5)-trisphosphate; PIP2: phosphatidylinositol 4,5-bisphosphate; PDK1: pyruvate dehydrogenase kinase 1; PTEN: phosphatase and tensin homolog; AKT: protein kinase B; Gsk3β: glycogen synthase kinase; mTORC-1: mammalian target of rapamycin; TSC1: hamartin; TSC2: tuberin; *Pdcd4*: programmed cell death 4 gene.

**Figure 7 biomedicines-10-03186-f007:**
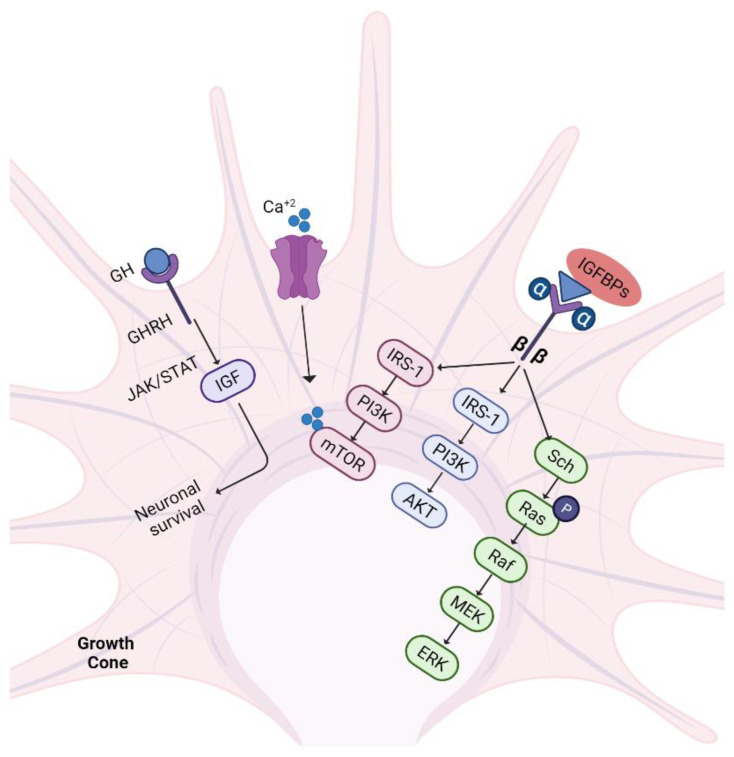
IGF-1/GH pathway. Nerve trauma triggers the release of IGF-1 at the injury site. IGF-1 binds to IGF-1R with the help of binding proteins and increases axonal outgrowth by acting in 3 different ways. Concomitantly, GH also activates IGF by JAK/STAT signalling and promotes neuronal survival. GH: growth hormone; GHRH: growth hormone receptor hormones; JAK/STAT: Janus kinase/signal transducer and activator of transcription; IGF: insulin-like growth factors; IGFBPs: IGF binding proteins; IRS-1: insulin receptor substrate-1; PI3K: phosphatidylinositol-3-kinase; AKT: protein kinase B; mTOR: mammalian target of rapamycin.

**Figure 8 biomedicines-10-03186-f008:**
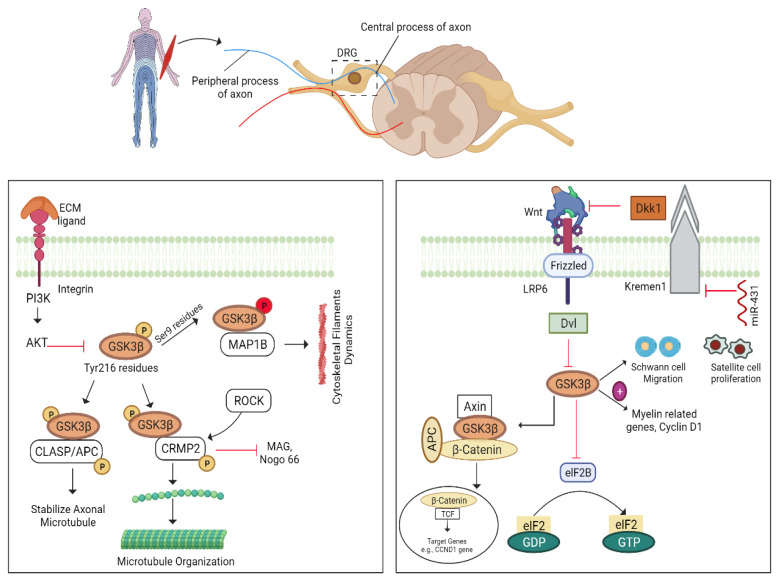
GSK3β mediated pathways. GSK3β mediates CLASP/APC signalling and Wnt signalling to stabilize growth cone and axon regeneration. ECM: extracellular matrix proteins; PI3K: phosphatidylinositol-3-kinase; AKT: protein kinase B; GSK3β: glycogen synthase kinase; CLASP/APC: clip-associated proteins/tumor suppressor protein; CRMP2: collapsin response mediator protein 2; *Map1b*: microtubule-associated proteins; ROCK: Rho-associated kinases; MAG/Nogo; myelin inhibitors; Wnt: protein; Dkk1: Dickkopf-1; Dvl: dishevelled proteins; GDP: guanosine diphosphate; GTP: guanosine triphosphate.

**Figure 9 biomedicines-10-03186-f009:**
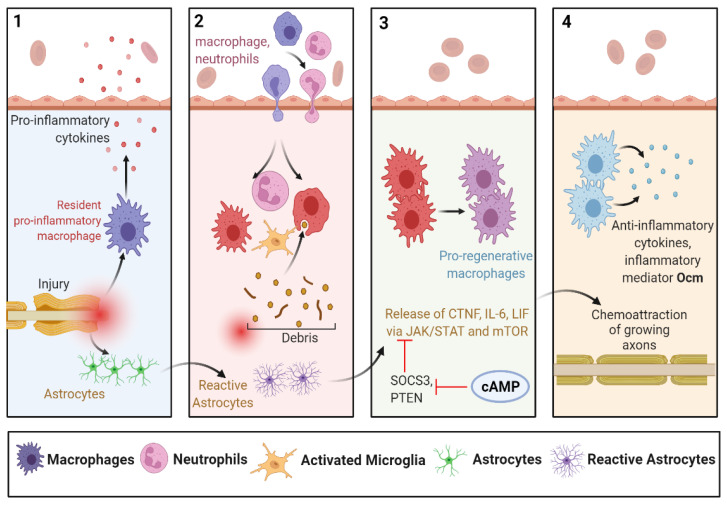
Regeneration by astrocytes/inflammatory mediators. Injury and inflammatory stimuli upon CNS contusion/trauma trigger different pathways and cytokines release. Migration of macrophages and neutrophils toward the injury site, as well as morphological changes of astrocytes promote axonal regeneration. IL-6: interleukin 6; LIF: leukemia inhibitory factor; CTNF: ciliary neurotrophic factor; cAMP: cyclic adenosine monophosphate: PTEN: phosphatase and tensin homolog; SOCS3: suppressor of cytokine signalling 3; Ocm: oncomodulin.

**Figure 10 biomedicines-10-03186-f010:**
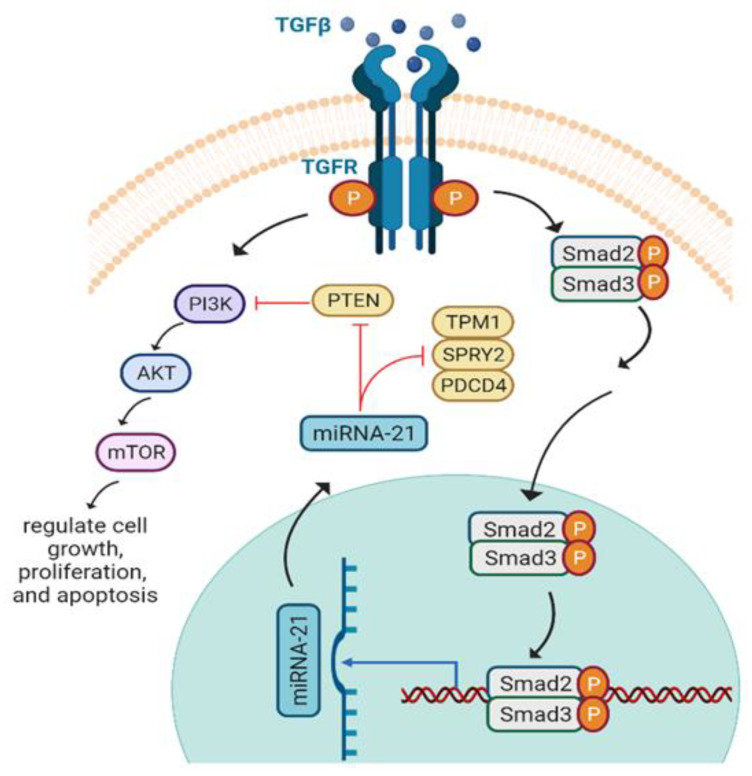
mi-RNA-21 signaling and axonal growth. Following trauma, TGF-β1 binds to particular TGF receptors and stimulates PI3K/AKT/mTOR signalling as well as phosphorylating and translocating SMAD2 and SMAD3 into the nucleus where it modulates miR-21 expression and allows it to move to the cytoplasm and inhibits growth suppressors such as PTEN, *Pdcd4*, *Spry2*, and *Tpm1* to regulate activation of astrocytes and axonal growth. TGFβ: transforming growth factor; TGFR: transforming growth factor receptors; Smad; PI3K: phosphatidylinositol-3-kinase; PTEN: phosphatase and tensin homolog; AKT: protein kinase B; mTOR: mammalian target of rapamycin; Pdcd4: programmed cell death 4 gene; Tpm1: tropomyosin 1; Spry2: sprouty RTK signalling antagonist 2.

**Figure 11 biomedicines-10-03186-f011:**
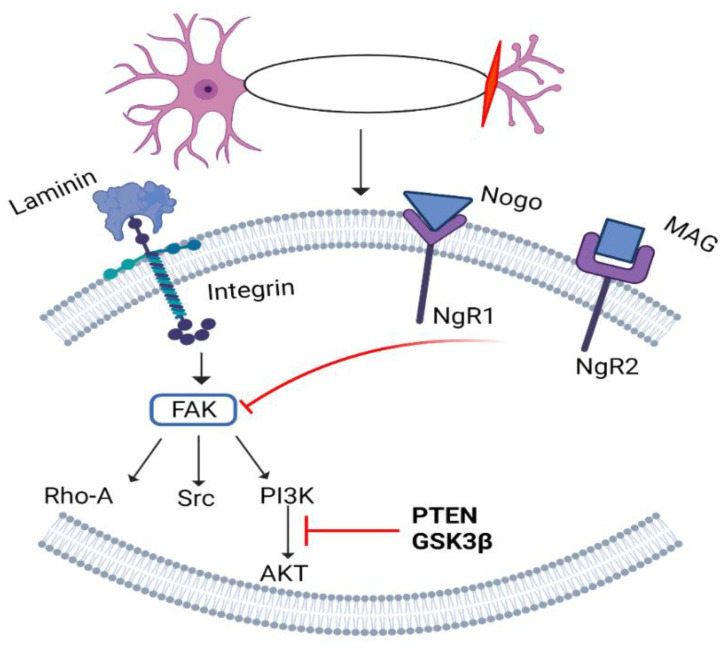
Integrin/FAK pathway. After nerve damage, integrin binds to different extracellular matrix ligands to activate FAK and PI3K/AKT, and Src. At the same time, Nogo and MAG not only bind to their respective receptors but also interfere with integrin to prevent activation of FAK. FAK: focal adhesion kinase; PI3K: phosphatidylinositol-3-kinase; PTEN: phosphatase and tensin homolog; AKT: protein kinase B; Gsk3β: glycogen synthase kinase; MAG: myelin-associated glycoproteins; NgR: Nogo receptor; Rho-A/Src: protein.

**Figure 12 biomedicines-10-03186-f012:**
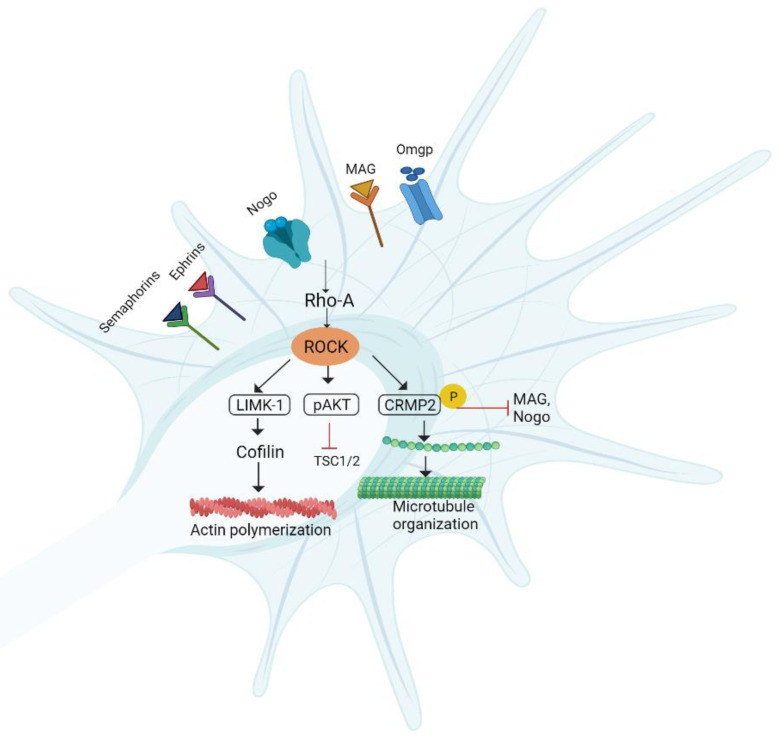
RhoA/ROCK/LIMK pathway. Different myelin inhibitors activate RhoA and Rho-associated kinases (ROCK), which in turn activate LIMK-1, cofilin, CRMP2, and AKT to control axonal development. RhoA and ROCK inactivation enhances cytoskeletal actin filament formation and prevents axonal degeneration to boost growth cone stability. TSC1/2 (hamartin–tuberin); AKT: protein kinase B; MAG: myelin-associated glycoproteins; Omgp: oligodendrocyte myelin glycoprotein; CRMP2: collapsin response mediator protein 2; LIMK: LIM domain kinase.

**Figure 13 biomedicines-10-03186-f013:**
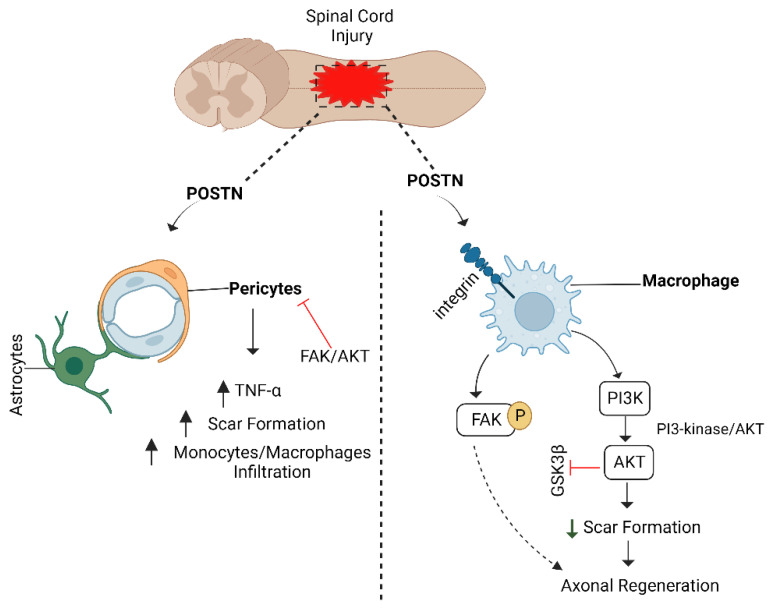
POSTN/integrin pathway. POSTN binds to pericytes and increases scar formation that hinders CNS neurons’ regeneration capacity. On the other side, POSTN binds to integrin receptors and activates PI3K/AKT and FAK signalling. AKT increases monocyte/macrophage migration at the injury site and prevents the formation of scars to increase axon regeneration in CNS neurons. POSTN: periostin; PI3K: phosphatidylinositol-3-kinase; AKT: protein kinase B; FAK: focal adhesion kinase; CNS: central nervous system.

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
