# Peer review of "Axonal Regeneration: Underlying Molecular Mechanisms and Potential Therapeutic Targets"

_biomedicines, 2022, doi:10.3390/biomedicines10123186_

Round 1

Reviewer 1 Report

In this review, the authors discuss the underlying molecular mechanisms for axonal regeneration and provide potential therapeutic targets for axonal injury and healing. The topic is interesting. However, it’s unclear what’s the different roles of those molecular pathways in the CNS and PNS. It’s better to compare them in terms of their contribution to axonal regeneration in the CNS and PNS. Additional comments are listed below.

1)      “RAGs” in Keywords should be spelled out.

2)      Lines 109-111: “HAT (histone acetyltransferases)” should be “histone acetyltransferases (HAT). “HDACs (Histone deacetylases)” should be “Histone deacetylases (HDACs)”.

3)      In Table 1, RGC and other abbreviations should be spelled out at the first time.

4)      Line 661: “SC” should be spelled out.

5)      In Section 6, what’s the meaning of the title “CNS injury signals”? Are inflammatory mediators, miRNA-21, integrin/FAK pathway, RhoA/ROCK/LIMK pathway, and POSTN/integrin signaling only involved in axonal regeneration in the CNS?

6)      Line 1118: what’s “invironment”? Is it “environment”?

7)      English grammar should be double-checked.

Reviewer 2 Report

The manuscipt is well written and comprehensive

Reviewer 3 Report

In the work "Axonal regeneration: underlying molecular mechanisms and potential therapeutic targets", Akram and coworkers tackle the thematic area of axonal regeneration. The authors first distinguish between the process occurring in the CNS and in the PNS. Then they enter in the details of the numerous molecular mechanisms taking part to the process, for each of which they provide pathway, targets, and possible pharmacological treatments.

The work appears as exhaustive, as also witnessed by the great number of figures, tables and references collected. However, some issues could be better tackled to improve the manuscript:

1) page 3 and throughout the text: RAGs and MAG and MAI (line 139) are poorly defined, details should be integrated along with reference specific for these targets

2) page 8 paragraph 5.4 neurotrophins: the authors acknowledge all the signalling effectors downstream TrK/p75 axis. However, signalling of neurotrophins is recongized to be distinguished in local signalling (at the tip) or distal signalling (at the soma, via gene expression regulation). The authors may try to distinguish between the signaling mechanisms occuring at these 2 locations during axon regeneration. Also, p75NTR is known to interact with the Nogo/MAG signaling axis, and this is not acknowledged in the manuscript.

3) page 14: table 1. At the mRNA section, three miRNA are listed. But miRNAs are not mRNAs. I am also a bit puzzled by the fact that epigenetic modifications consitute an initial paragraph of the review, but then the action of miRNA (epigenetic regulators) is listed in the molecular signalling paragraph.

4) the paper is full of typos, english grammar errors, full stops missing or duplicated etc. This point should be regarded as a major point and I would recommend that the paper is checked by a native english speaker. Also, many figures result cut and not clearly visible in the pdf version which was uploaded for review. the authors should check this aspect. For example, in the abstract, line 34 an "are" should be removed, line 227 "upregulate" and "regulate" need an "s" at the end and so on.

5) the last part of the abstract and of the conclusion looks weak. what does "in clinical investigations, these molecular targets must be manipulated to ensure functional restoration of wounded axons" mean? The authors should try to be more precise and pragmatic. How far are the pharmacological treatments proposed throughout the text far from the clinic for a use in axonal pathologies and why?

Round 2

Reviewer 1 Report

The authors have addressed my comments.

Reviewer 3 Report

congratulations to the authors for the high quality of the revision work.

in my opinion the paper is suitable for publication.